# Ultrasound Evaluation of Onset Core Muscle Activity in Subjects with Non-Specific Lower Back Pain and Without Lower Back Pain: An Observational Case–Control Study

**DOI:** 10.3390/diagnostics14202310

**Published:** 2024-10-17

**Authors:** María Cervera-Cano, David Valcárcel-Linares, Samuel Fernández-Carnero, Luis López-González, Irene Lázaro-Navas, Daniel Pecos-Martin

**Affiliations:** 1Universidad de Alcalá, Facultad de Medicina y Ciencias de la Salud, Departamento de Enfermería y Fisioterapia, Grupo de Investigación en Fisioterapia y Dolor, 28801 Alcalá de Henares, Spain; fisioterapiacervera@gmail.com (M.C.-C.); david.valcarcel.1994@gmail.com (D.V.-L.); luislopezgonzalez4@gmail.com (L.L.-G.); daniel.pecos@uah.es (D.P.-M.); 2Department of Clinical Trials, Hospital 12 de Octubre, 28041 Madrid, Spain; 3Technical Support Office for European Programs, Fundación para la Investigación e Innovación Biosanitaria en Atención Primaria (FIIBAP), 28003 Madrid, Spain; 4Physiotherapy Service, Ramón y Cajal Hospital, 28034 Madrid, Spain; ireneln88@gmail.com

**Keywords:** non-specific lower back pain (NLBP), rehabilitative ultrasound imaging (RUSI), transversus abdominis, diaphragm function, pelvic floor muscles

## Abstract

Lower back pain (LBP) has been the leading cause of disability since 1990. **Objectives**: The main objective of this observational case–control study was to evaluate, using ultrasound, whether there were differences in the onset and ratio of core muscle contraction between subjects with non-specific chronic lower back pain and healthy subjects. **Methods**: A total of 60 participants (52% women), split between those with non-specific chronic lower back pain (*n* = 26) and healthy (*n* = 34) subjects, were recruited. Initial muscle contraction of the lateral abdominal wall, pelvic floor, lumbar multifidus, and respiratory diaphragm was measured using ultrasound. The abdominal drawing-in maneuver, contralateral arm elevation, the Valsalva maneuver, and voluntary contraction of the pelvic floor in seated and standing positions were performed. The muscle thickness of the lateral abdominal wall and lumbar multifidus and excursion of the pelvic floor and diaphragm at rest and during testing were also analyzed. **Results**: No differences were found between the groups in the initial contraction. Statistically significant differences were found in the following variables: diaphragm excursion (*p* = 0.032, r = 0.277) and lumbar multifidus ratio (*p* = 0.010, r = 0.333) in the standing–abdominal retraction maneuver; pelvic floor excursion (*p* = 0.012, r = 0.325) in the standing–contralateral arm raise; and transverse abdominis ratio (*p* = 0.033, r = 0.275) in the sitting–contralateral arm raise. A statistically significant interaction between the groups and body mass index was observed in resting diaphragm excursion (*p* = 0.018, partial eta squared = 0.096) during sitting–voluntary pelvic floor contraction. **Conclusions**: It cannot be concluded that there is a specific pattern of core activation in any of the groups. However, statistically significant differences were found in the contraction indexes of the lumbopelvic musculature.

## 1. Introduction

Since 1990, lower back pain (LBP) has been the leading cause of disability [1]. In 85–90% of cases, the exact source of pain cannot be precisely identified, and patients are diagnosed with non-specific lower back pain (NSLBP) [1]. LBP affects 18% of adults and has a significant socioeconomic impact on a global scale [2]. In 10% of cases, LBP becomes chronic (CNSLBP) when it persists for more than twelve weeks in one year [1,2]. Factors such as occupational strain, obesity, and unhealthy lifestyle habits have been identified as increasing the risk of developing NSLBP [2].

Additionally, psychological and social factors, such as depression, anxiety, catastrophizing, and low self-efficacy, have been proposed to be contributors to the persistence of pain [3]. However, the involvement of musculoskeletal factors remains ambiguous.

Although there is no universally accepted definition of the “core”, it generally refers to the abdominal–pelvic functional unit, which includes not only vertebral segments and their associated passive supporting structures or the neural system but also four key pillars: the lumbar multifidus (LM), the lateral abdominal wall (LAW), the respiratory diaphragm (DPH), and the pelvic floor (PF) [4,5]. Together, these muscles form a cylindrical system that operates in harmony to regulate lumbopelvic forces, providing trunk stability, improved control, movement efficiency, balance, coordination, and enhanced motor control (including postural alignment and firmness). However, while the term “core” refers to multiple muscle groups, it is more functional in nature than anatomical [4,5].

Several techniques have been employed to evaluate muscle function and/or core structure, with electromyography (EMG) considered the “gold standard”. More recently, rehabilitative ultrasound imaging (RUSI) has emerged as a reliable alternative to magnetic resonance imaging (MRI) and has been shown to be comparable to the “gold standard” of EMG or MRI [6].

Additionally, EMG has shown certain limitations when evaluating muscle behavior [7]. Ultrasound has been recognized as a reliable and safe alternative tool for assessing musculature [6]. The application of the motion mode (M-Mode) in ultrasound has proven effective in evaluating muscle contractions (onset) in individuals with NSLBP [4,8].

Many researchers have explored the musculoskeletal causes of NSLBP, particularly focusing on movement alterations, muscle activity changes, and/or muscle atrophy differences when compared to asymptomatic individuals [9]. These variations seem to play a key role in the assessment and management of patients with LBP [6].

In a recently published systematic review, it was concluded that there are few studies that analyze the global behavior of the core beyond the lateral abdominal muscle complex (TRA, IO, and EO). In addition, none of them involve examining overall muscle behavior, including other important muscle components within the core function, such as the pelvic floor or diaphragm [4]. It was concluded that one of the main limitations in conducting ultrasound studies on the core’s functional unit is the absence of a measurement tool capable of supporting multiple probes simultaneously [4].

Regarding the concept of the core and motor control, it is understood that it is not only the individual contraction and structure of the muscles that are important but also the coordination between the various muscle groups that comprise this functional unit [10]. Consequently, research that examines the overall muscle behavior of the core and its structure in the context of pain compared to healthy individuals would provide valuable insights into the potential differences between these groups and their possible role in NSLBP.

In the present study, we hypothesized that any deviation in the timing, synergy, and overall behavior of this functional unit could potentially contribute to the persistence or onset of CNSLBP. The exercises of abdominal drawing-in maneuver) [11], contralateral arm lift (CAL) [12], voluntary pelvic floor contraction (VPFC) [13] and the Valsalva maneuver [14], were selected to elicit the specific activation of the muscles in the abdominal–pelvic region and observe their interaction with other muscle groups. The ADIM was designed to facilitate the coactivation of the TrA and LM to stabilize the trunk before limb movement; the CAL maneuver is aimed at identifying LM muscle activity and size during test execution; and the VPFC was used to assess pelvic floor muscle function and contraction through bladder displacement. Lastly, the Valsalva maneuver was employed to examine diaphragm behavior, as it involves a breathing pattern consisting of forced exhalation with a closed glottis to increase spinal stability by raising intra-abdominal pressure, activating the entire abdominal musculature. The assessment of these maneuvers in both standing and sitting positions in ultrasound studies focused on lower back pain is justified by their relevance to daily and occupational activities, in addition to their propensity to induce or exacerbate lower back pain symptoms [15,16,17].

The primary objective of this observational study was to understand the onset of muscle contractions in patients with NSLBP and healthy subjects. Additionally, as a secondary outcome, in this study, we aimed to analyze the ratio between rest and contraction/activation, as well as rest and excursion, for each muscle group in both subject groups. These variables were chosen based on their well-established association in the existing literature with lumbopelvic stability and core function, rendering them essential for this investigation [11]. Moreover, these factors have been identified as critical predictors of pain and musculoskeletal function [11], further justifying their selection as primary research objectives.

## 2. Materials and Methods

The present study was an observational case–control study. The study was developed according to Strengthening the Reporting of Observational Studies in Epidemiology (STROBE) guidelines [18]. Data were collected between December 2021 and February 2023. The study was conducted according to the guidelines of the Declaration of Helsinki and approved by the Institutional Review Board (or Ethics Committee) of the Ethical Committee for Research and Animal Experimentation (CEIM) of the University of Alcalá (CEIM/HU/2019/41). The protocol for this study has also been published [19].

### 2.1. Participants

The participants included individuals with CNSLBP and asymptomatic individuals, selected based on specific criteria and recruited through non-probability convenience sampling. The participants were contacted by the University of Alcalá in Madrid. They were provided with the “Adult Research Patient Information Sheet”, an informed consent form, and a “Participant Data Collection Form”. A physician at their healthcare center diagnosed the individuals with CNSLBP. After signing the informed consent and completing the data collection form, a sub-investigator (D.V.-L. or L.L.-G.) reviewed the inclusion and exclusion criteria. The participants were then assigned to either the CNSLBP (case) or control group.

### 2.2. Selection Criteria

For the chronic CNSLBP group, the following inclusion criteria were applied:Individuals with CNSLBP as defined by the scientific literature [1,2];Participants diagnosed by a physician at their healthcare center;Adults aged between 18 and 60 years [14,20];A history of NSLBP for at least twelve weeks in the past year [2];A pain score of 3 or higher on a 10-point Visual Analog Scale (VAS) [21,22].

The control group consisted of asymptomatic individuals aged 18 to 60 years old [14] who had not experienced lower back pain at the time of the assessment or in the past year.

Exclusion criteria included general surgical intervention, vertebral pathology (such as fractures, cancer, or infection), inability to stand or sit, or inability to perform the required maneuvers. Pregnancy was also an exclusion criterion [20].

### 2.3. Sample Size Calculation

The sample size was calculated using the software G*Power version 3.1.9.6 © (Düsseldorf, Germany) [23]. An effect size of d = 0.75, an alpha error of 0.05, a power of 80%, and an allocation ratio of 1.3 were estimated. Ultimately, the total sample size was 60 subjects (cases: *n* = 24; controls: *n* = 36).

### 2.4. Outcomes

The main outcome measured was the timing of muscle contractions in the four core muscle groups during various tasks, such as the abdominal drawing-in maneuver (ADIM), contralateral arm lift (CAL), the Valsalva maneuver, and voluntary contraction of the pelvic floor, in both sitting and standing positions:Timing of muscle contraction in the lateral abdominal wall (LAW), including the transversus abdominis (TrA), internal oblique (IO), and external oblique (EO).Timing of muscle contraction in the lumbar multifidus (LM).Movement onset of the diaphragm (DPH).Movement onset of the pelvic floor (PF).

The onset of muscle activation was defined as the moment when a muscle begins to contract in response to a stimulus or command [7]. Based on the onset of activation, an activation pattern was established for each maneuver in each position for both groups of subjects.

As a secondary outcome, differences were measured in the ratios (contraction/relaxation) of the LM and LAW (IO, EO, and TrA), in addition to during excursion upon inspiration and the resting state of the respiratory diaphragm and the pelvic floor musculature during both seated and standing maneuvers. These measurements were analyzed using the following formula: [(Muscle thickness or excursion during test − Muscle thickness or excursion at rest)/Muscle thickness or excursion at rest] × 100. Diaphragmatic excursion was examined by comparing differences in inspiration before and during the maneuver; in comparison, pelvic floor contraction was visualized through bladder displacement. Lastly, the degree of disability was assessed using the Oswestry disability questionnaire [24].

### 2.5. Procedures

The data were gathered by a skilled physiotherapist specialized in RUSI, with 10 years of experience (S.F.-C.), together with the principal investigator (M.C.-C.), who had completed 60 h of ultrasound training [25]. Both researchers (S.F.-C. and M.C.-C.) were unaware of the participants’ group assignments while collecting the data. Two sub-investigators (D.V.-L. and L.L.-G.) handled the participants’ data collection forms and assigned them to either the case or control group.

Once the participant completed all the information and was assigned to the corresponding group, all the documentation was filed in the investigator’s binder to prevent the principal investigator from inadvertently breaking the blind.

Four wireless US probes (Sonostar Technology Co., Ltd. © (Guangzhou, China)), two linear probes with a bandwidth of 7.5–10 MHz, and two convex probes with a bandwidth of 3.5–5 MHz were used. The probes were secured using a custom-designed fixation belt created by the research team, which has been registered as a utility model with the Spanish Patent and Trademark Office (Oficina Española de Patentes y Marcas (OEPM)) with patent number U202131486 (Figure 1) [26].The customized fixation belt facilitated the simultaneous collection of ultrasound images and videos during the maneuvers.

Three of the probes, corresponding to the LAW, LM, and DPH, were positioned on the patient’s right side to utilize the liver’s sound-amplifying properties for measuring DPH movement [27]. The fourth probe, corresponding to the PF, was placed transversely and suprapubically along the midline [28]. The probe placements were based on validated measurement areas for each muscle group, as established by various researchers [27,28,29,30].

All four ultrasound images were displayed and synchronized on a computer using a custom video capture card (Blackmagic Design DeckLink Quad HDMI Recorder, Port Melbourne, Victoria, Australia), facilitating the analysis of muscle contraction onset in addition to the thickness and movement of the muscles.

Ultrasound analysis was conducted to assess the simultaneous and individual muscle contraction onset of the four core muscle groups, in addition to their thickness/movement, during the abdominal drawing-in maneuver (ADIM), contralateral arm lift (CAL), voluntary pelvic floor contraction, and Valsalva maneuver, both in sitting and standing positions. These positions were chosen because they are considered potential risk factors for the development and persistence of NSLBP [31].

To avoid bias, the order of the maneuvers and positions was randomized using opaque envelopes assigned to each participant. All tests were performed once in both sitting and standing positions, with a two-minute rest between maneuvers. The researchers determined that a two-minute rest between each maneuver and position, given the low intensity of the tests, would be sufficient to prevent fatigue. Each maneuver was explained to the participants before measurements were taken.

Abdominal Drawing-in Maneuver (ADIM)

The ADIM is intended to activate both the TrA and LM, providing trunk stabilization prior to limb movement [11]. Participants were trained using standard tactile and verbal instructions while in sitting and standing positions before measurement [11]. For the ADIM, participants were instructed to “Take a relaxed breath in and out, then draw in the lower abdomen without moving the spine” [30].

Contralateral Arm Lift (CAL)

The CAL maneuver assesses the LM’s muscle activity and size during the test. Prior to measurement, the subjects were asked to breathe normally before and during the maneuver. With the arm fully straightened and the wrist in a neutral alignment, the participant lifted the arm to shoulder level and maintained the position for 2 s [12].

Voluntary Pelvic Floor Contraction (VPFC)

To assess the function and contraction of the PF musculature, transabdominal ultrasound was used with the subject placed in the supine position [13]. Prior to measurement, the subjects were asked to breathe normally before and during the maneuver. Upon reaching the optimal measuring position, the participant was instructed to activate their pelvic floor muscles with the command: “try to close the urethra as if you wanted to hold back urine”. The contraction of the pelvic floor was maintained for 6–10 s [32]. The aim was to measure the displacement of the bladder base during voluntary contraction of the pelvic floor muscles after drinking 500 to 750 mL of water for adequate visualization. The measurements were performed using the M Mode since it facilitated observation of the contraction of the PF musculature from the initial moment when its contractile activity began to occur [13].

Valsalva maneuver

The Valsalva maneuver is a breathing technique involving forceful exhalation with the glottis closed to increase spinal stability by raising intra-abdominal pressure, as it activates the entire abdominal musculature [14]. The instructions provided to the participants for performing the Valsalva maneuver were as follows: (1) “Take air into the ears with the mouth and nostrils closed”, and (2) “Apply pressure towards the pelvic floor, as if trying to defecate”. No specific instructions were given regarding the contraction or relaxation of the abdominal muscles or the pelvic floor. Muscle activity was recorded within 3 s of the participant’s maximum contraction [14].

The maneuvers and their characteristics are summarized in Table 1 and Table 2.

The US measurement of each muscle group in activation and rest can be shown in Figure 2.

### 2.6. Data Processing

The data used in this study were collected from December 2021 to February 2023. First, a participant information sheet providing details on the observational study was provided. Once the participant had carefully read the information about the study and agreed to participate, all participants signed two copies of an informed consent form (ICF), with one copy retained by the participant and the other retained by the research team. The Participant Data Collection Form, which includes socio-demographic information, such as height, weight, body mass index (BMI), and personal medical history, was gathered from all participants. Only those assigned to the case group completed the Visual Analog Scale (VAS) [22] and the Oswestry Disability Index scale [24]. To ensure data authenticity, all of the participants filled out the questionnaires independently.

After completing all of the forms, each participant was fitted with a customized fixation belt and the corresponding ultrasound (US) probes (Figure 3). Using the US M-Mode, the initiation of movement for each specific muscle group was identified. These findings were compared with the characteristics of each muscle group, including the activation/contraction ratio of the lumbar multifidus (LM) and the lateral abdominal wall (LAW), in addition to the excursions of the endopelvic and respiratory diaphragms.

The US videos and images of each probe were displayed on four tablets (iPad^®^ 7th generation, Apple Inc. ©, Los Altos, CA, USA) using the USG Wireless program WLUSG TP ^®^ software (v. 3.5.0, SonoStar Technology Co., Ltd. ©, Guangzhou, China). These tablets were connected to a computer (MacBook Pro ^®^ 2019, Apple Inc. ©, Los Altos, CA, USA) through a customized video capture card (Blackmagic Design DeckLink Quad HDMI Recorder) (Figure 4).

To measure the onset of muscle contraction and activation for each muscle group, OBS (Open Broadcaster Software ^®^ V.28.1.2 Boston, MA, USA) was utilized. This software enabled the simultaneous real-time visualization of four videos, making it easier to observe both specific and global muscle contractions. Additionally, a stopwatch was used to mark the onset of muscle contraction. The collected data were then saved and imported to a laptop for subsequent analysis by a blinded examiner using the Free software FIJI ^®^ (V.2.9.0 Cambridge, UK), which showed an ICC between 0.78 and 0.99 [35,36] (Figure 5).

### 2.7. Statistical Analysis

The Shapiro–Wilk test was used to assess the normality assumption for each group. For the descriptive analysis of continuous quantitative variables, the mean and standard deviation (SD) were reported when normality was met, whereas the median and the first and third quartiles were used otherwise. For nominal variables, absolute and relative frequencies were provided. Group homogeneity in demographic variables was evaluated using the unpaired *t*-test for continuous variables that met the normality assumption or the Mann–Whitney U test for those that did not. The Pearson chi-square (χ^2^) test was employed for nominal variables.

To validate the customized fixation belt measurements, an inter-rater reliability analysis was performed on 10 subjects. First, the researcher S.F.C. placed the belt on the participant, took a photograph using the four probes, and removed the belt. After 5 min, the researcher M.C.-C. repeated the procedure. The analysis was carried out under the assumption of a two-way mixed effects model and absolute agreement, using the average of 3 measurements (muscle thickness and excursion at rest in each muscle) through ICC (3, 3) [37]. The standard error of measurement (SEM = SD × √ (1-ICC)) and the minimum detectable difference (MDD95 = SEM × √2 × 1.96) were calculated. The Bland–Altman agreement charts were produced, estimating the agreement limits as the mean difference ±1.96 × SD [30].

The Pearson χ^2^ test was carried out for the analysis of the differences between independent variables (cases and controls) and the onset muscular contraction (nominal variable) of each muscle in every maneuver and position. A number from 0 to 40 denoted each activation pattern. The effect size was estimated using the contingency coefficient (φ2 = χ^2^/N).

For the analysis of the differences between the groups and the muscle ratio in each maneuver (ADIM, CAL, Valsalva, and VPFC) and position (sitting or standing), the unpaired *t*-test and Cohen’s d were used, or the Mann–Whitney U test and Rosenthal’s r were used if the former could not be used. When the outcomes were normally distributed, covariance analysis (ANCOVA) was used, with the ratio as the intra-subject factor, the group (pain or healthy) as the inter-subject factor, and BMI as a covariant. The partial eta squared (η2p) was used as an effect size estimator of the ANCOVA.

All analyses were carried out with Statistical Package for the Social Sciences software (SPSS^®^) version 24 for Mac. An α level of 0.05 with a 95% confidence interval (CI) was assumed for all analyses.

## 3. Results

Sixty-six participants were screened for inclusion in the study. Six participants were excluded as the images were of poor quality and deemed unreadable [4], and an additional two participants were unable to complete the measurement tests due to dizziness during the maneuvers. These data are illustrated in the flow diagram in Figure 6. 

A total of 60 participants (52% women) were recruited and analyzed in the study. Sample recruitment was carried out between December 2021 to February 2023. Subjects’ characteristics and baseline scores by allocation group are summarized in Table 3.

### 3.1. Body Mass Index Results

BMI was found to be statistically different between to two groups (*p* = 0.023).

### 3.2. Inter-Rater Reliability Analysis

The inter-rater reliability analysis of the muscle measurements with the customized fixation belt showed good reliability for DPH excursion (ICC = 0.889) and excellent reliability for PF excursion (ICC = 0.989), EO (ICC = 0.982), IO (ICC = 0.966), TrA (ICC = 0.988), and LM (ICC = 0.983) thicknesses (Appendix A).

### 3.3. Bias

Bland–Altman plots showed agreement between measurements, and no systematic bias was found (Appendix A). 

### 3.4. CORE Activation Pattern in Subjects with and Without Chronic Lower Back Pain

Forty-one different activation patterns were identified, distributed among cases and controls during the performance of the four maneuvers (ADIM, CAL, Valsalva, and VPFC) while seated or standing. These patterns were named in ascending order of appearance (Appendix A). In some of the maneuvers, activation of a muscle group was not detected (-). No statistically significant results were found for a specific activation pattern for each maneuver (ADIM, CAL, Valsalva, and VPFC), neither in standing nor in seated position in the group of subjects with CNSLBP nor in the group of healthy subjects.

The results for the mean differences between cases and controls for each maneuver in standing and seated positions for which the outcome variable is the activation pattern can be found in the Appendix A. Activation patterns with higher frequency in each maneuver and position are shown in Table 4.

### 3.5. Muscle Thickness in Subjects with and Without Chronic Lower Back Pain

Regarding muscle thickness, a statistically significant difference was found in DPH excursion (*p* = 0.032, r = 0.277) and LM ratio (*p* = 0.010, r = 0.333) during the standing—ADIM; in PF excursion (*p* = 0.012, r = 0.325) during the standing—CAL; and in the TRA ratio (*p* = 0.033, r = 0.275) during the sitting—CAL (Appendix A).

Of note, the ANCOVA results showed a statistically significant interaction between the groups and BMI in DPH excursion at rest (*p* = 0.018, η2p = 0.096) during sitting—VPFC.

## 4. Discussion

The main objective of this study was to assess the activation of the four muscle groups (lateral abdominal wall, pelvic floor, diaphragm, and lumbar multifidus) during four maneuvers in both standing and sitting positions in subjects with chronic lower back pain and healthy subjects simultaneously. As a secondary objective, potential differences in the contraction ratio between the aforementioned muscles were investigated, and variations in diaphragm excursion and resting state, in addition to the pelvic floor musculature, were analyzed in each maneuver and position. These variables have been identified as critical predictors of pain and lumbopelvic stability and function [11]. In this study, the hypothesis was raised that alterations in the intermuscular synergy and the overall behavior of the abdominopelvic unit could be related to the perpetuation or cause of CNSLBP.

The results show that no differences were found in the activation patterns of the abdominopelvic musculature between subjects with CNSLBP and healthy subjects. However, greater excursion of the diaphragm and a higher ratio in the lumbar multifidus were found in subjects with pain during the ADIM in the standing position. Additionally, the pelvic floor exhibited greater excursion (greater bladder displacement) in cases during the CAL maneuver in the standing position, in addition to a lower TrA contraction ratio when performing the same maneuver in the sitting position, compared to the control group.

The high variability found in activation patterns between both groups does not support that the sequence of core activation is implicated in CNSLBP. However, the changes found in muscle thickness and excursion of the muscle groups in response to functional demands, such as the ADIM and CAL maneuvers, suggest the potential involvement of the core and upper limb movements in this population.

Although ultrasound, especially the M Mode (M-Mode), has served as a reliable and safe alternative measurement tool for assessing this musculature in individuals with NSLBP [4,6,8], difficulties have been encountered in conducting simultaneous core evaluations, perhaps due to the absence of devices that enable such studies. In this study, we present a novel system for simultaneous CORE assessment using a restraining belt, which proved to be reliable and enables the evaluation of functional maneuvers or movements without operator dependence in all subjects.

The results of recent studies have revealed significant differences in morphology and muscle activation between subjects with chronic lower back pain (CLBP) and healthy individuals [14,15,38,39]. For example, it has been observed that, in subjects with CLBP, the thickness of the External Oblique (EO) in the sitting position is lower compared to other core muscle components, such as pelvic floor thickness and the diaphragm [14]. Furthermore, the thickness of the deep parts of the transversus abdominis (TrA) and the lumbar multifidus (LM) also decreases in subjects with CLBP compared to healthy subjects [14]. The results of this study support the potential involvement of movement or functional demand in the presentation of CLBP, as no changes were found in thickness and excursion at rest.

The results of other studies have highlighted the importance of the relationship between different core muscle groups, such as the TrA, EO, and ML, suggesting synergy among them and correlating ML fatigue with the thickness of these muscles in subjects with CLBP [40,41,42]. In this study, unlike ML in the standing position, TrA exhibited a lower contraction ratio in the sitting position during the ADIM in subjects with CLBP, which could explain the delayed activation described in previous studies [14,15,38,39]. This relationship and synergy between ML and TrA may differ in individuals with pain depending on whether the maneuver is performed in a standing or sitting position, which could have significant implications for the design of rehabilitation programs.

Atrophy of the ML in subjects with NSLBP has been described in several studies [29,43]. As a significant finding in this study, a higher contraction ratio was observed in subjects with pain compared to healthy individuals in the standing position and during the ADIM, which had not been previously described. It is suggested that, despite this musculature being primarily composed of tonic type S muscle fibers (slow), subjects with NSLBP may have a greater predominance of fast-fatigable type FF fibers after chronicity of lower back pain [44].

Conversely, the results of previous studies show a decrease in the thickness and excursion of the respiratory diaphragm in subjects with NSLBP compared to healthy individuals [27]. However, in the present study, the participants showed an increase in diaphragmatic excursion during inspiration while performing the ADIM in the standing position compared to healthy subjects. This significant result could be interpreted as subjects with CNSLBP tending to take deeper breaths before and during certain maneuvers, suggesting a different respiratory response between the two groups. This difference could be marked by the type of breathing, which could be compensatory, either apical or abdominal, due to pain or lack of activation of certain muscle groups. In fact, these results have already been noted in a study in which an initial relationship between hypocapnia (exaggerated respiration or ventilation in excess of metabolic needs) and increased modulation of TrA thickness during resting respiration using ultrasound was demonstrated [45].

Furthermore, a lack of a local activation strategy of the pelvic floor musculature has been described in women with CNSLBP. However, performing pelvic floor contraction during certain maneuvers may facilitate an increase in transversus abdominis thickness [46]. However, these results did not show statistical significance in this observational study, although increased activation of the pelvic floor was observed in subjects with CNSLBP during the CAL maneuver in the standing position.

A significant finding from this observational study is the notable heterogeneity in the results of muscle activation patterns, with 41 different patterns identified, none of which reached statistical significance. No characteristic activation pattern was identified in subjects with CNSLBP or in healthy subjects. Additionally, there were minimal statistically significant differences in the contraction/excursion ratio between healthy subjects and those with CNSLBP, contrary to the results of previous studies.

Another significant result was the relationship between a higher BMI in the participants, with this variable being statistically significant in resting DPH excursion (*p* = 0.018, η2p = 0.096) in the sitting position—VPFC, suggesting that BMI may moderately influence diaphragm excursion at rest. This assumption requires further analysis in subsequent studies in order to confirm this hypothesis.

Regarding limitations, the study design only allows for the establishment of potential causal hypotheses, which could be confirmed in prospective longitudinal studies, such as cohort studies, where increased diaphragm excursion, higher ML ratio, greater SP excursion, and/or lower TrA ratio are established as risk factors. Conversely, although EMG was not used as the “gold standard” for muscle behavior evaluation, the examiners have the necessary experience to perform muscular ultrasound measurements according to procedures outlined in the literature [25], and the measurement belt demonstrated excellent reliability. Lastly, despite the study’s moderate effect size (d = 0.706 after conversion using the formula d = 2r / √1 − r2) [47], it did not reach the estimated effect in the sample size calculation (d = 0.750). This discrepancy could be addressed by increasing the sample size.

## 5. Conclusions

In conclusion, no association was found between changes in core muscle thickness and a specific activation pattern in subjects with CNSLBP or in healthy subjects. These findings underscore the need for further research to better understand the role of the core musculature in the etiology and treatment of chronic lower back pain. In addition, they suggest that the cause of CNSLBP may be multifactorial, and the relationship between muscle activation of the core muscles and lower back pain remains undetermined. Therefore, additional studies with a larger sample size and involving different methods of simultaneous evaluation (EMG, MRI, etc.) are necessary to determine the role of the core musculature in the etiology of CNSLBP.

## Figures and Tables

**Figure 1 diagnostics-14-02310-f001:**
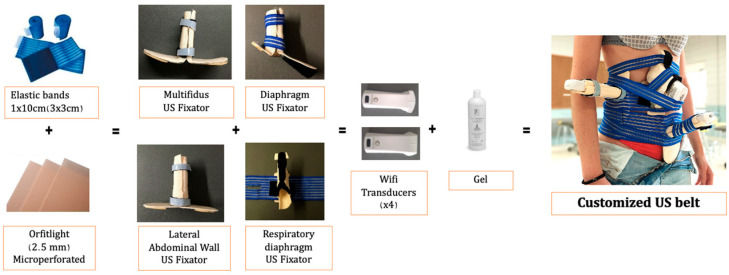
Customized fixation belt No. U202131486 (OEPM; Oficina Española de Patentes y Marcas).

**Figure 2 diagnostics-14-02310-f002:**
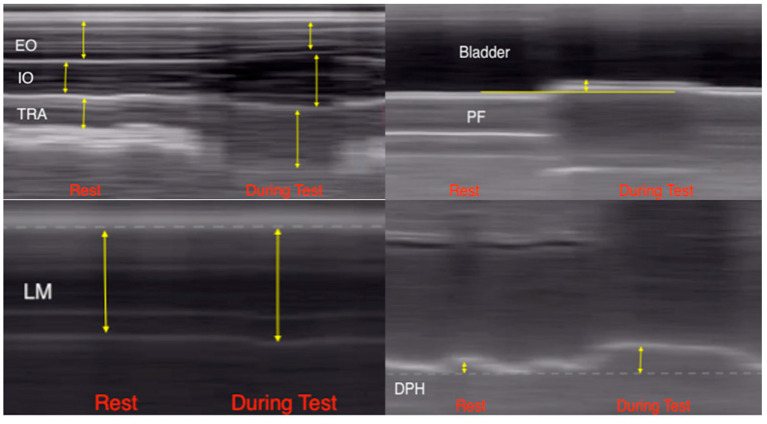
Caliper US placement. EO: external oblique; IO: internal oblique; TRA: transversus abdominis; LM: lumbar multifidus; PF: pelvic floor; DPH: diaphragm.

**Figure 3 diagnostics-14-02310-f003:**
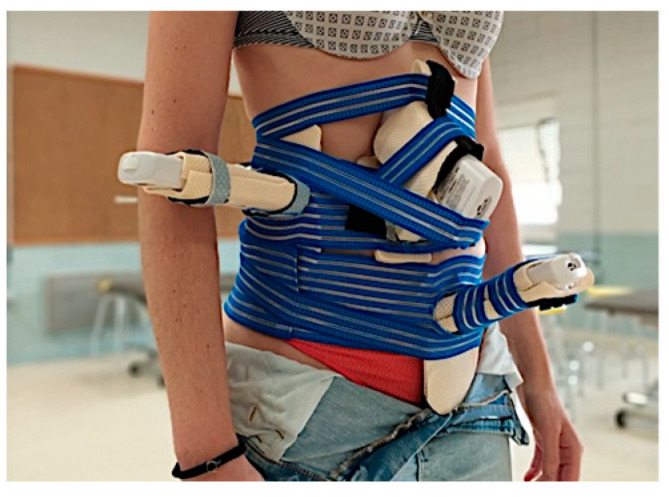
Customized fixation belt placed onto the subject.

**Figure 4 diagnostics-14-02310-f004:**
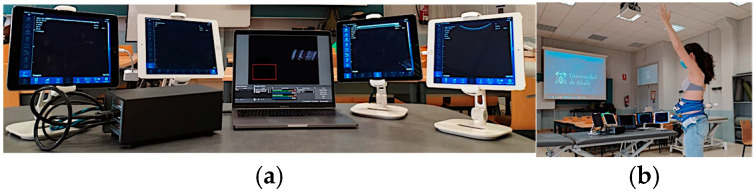
Ultrasound image visualization on a computer using a customized video capture card. (**a**) Four tablets with a customized video capture card and computer. (**b**) Four tablets with a customized video capture card and computer and the customized fixation belt placed onto the subject.

**Figure 5 diagnostics-14-02310-f005:**
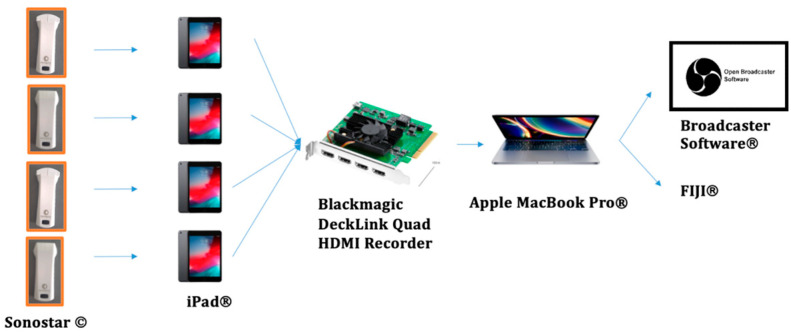
Ultrasound data processing.

**Figure 6 diagnostics-14-02310-f006:**
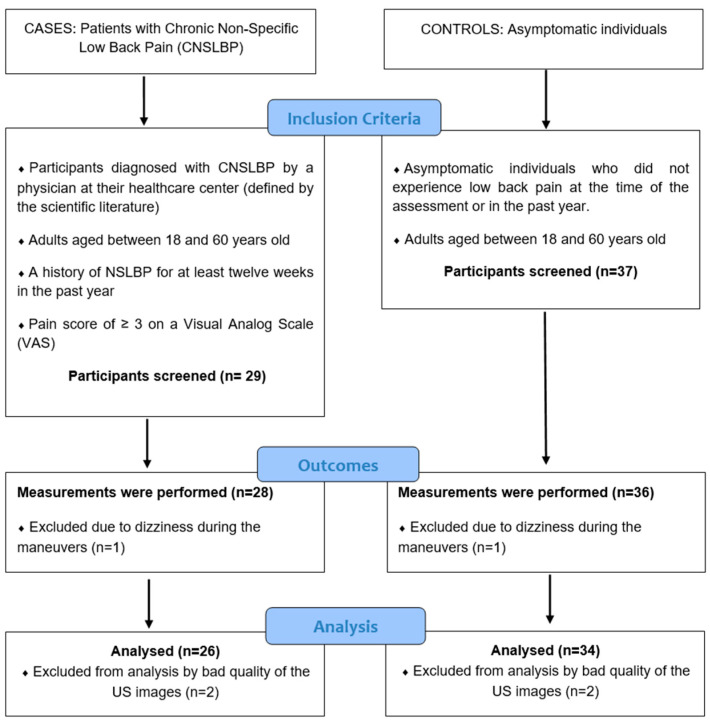
Flow diagram of the study design.

**Table 1 diagnostics-14-02310-t001:** Maneuvers and their characteristics.

Maneuver	Verbal Command	Muscle Contraction	Rest Between Maneuvers
ADIM [11]	“Take a relaxed breath in and out, hold the breath out, and then draw in the lower abdomen without moving their spine”	At the end of the exhalation	2 min
CAL [12]	“Raise the extended arm to shoulder height and hold it for two seconds. Then lower it naturally”.	2 s	2 min
VPFC [13]	“Try to close the urethra as if you wanted to hold back urine”	6–10 s	2 min
Valsalva [14]	“Apply pressure towards the pelvic floor, as if they intended to defecate”	3 s	2 min

ADIM: abdominal drawing-in maneuver; CAL: contralateral arm lift; VPFC: voluntary pelvic floor contraction.

**Table 2 diagnostics-14-02310-t002:** Summary of the methodology of the US evaluations.

Muscle	Probe	Transducer Placement	Transducer Bandwidth	Caliper Placing
DPH [27]	Convex	Right anterior axillary line, subcostal. The gallbladder and inferior vena cava were taken as anatomical landmarks.	2.5–3.5 MHz	Distance (mm) from the most distal apex of the right hemi diaphragm to the midpoint of the convex probe coinciding with the hyperechogenic curved line of the diaphragm dome.
PF [32]	Convex	Suprapubic, in the mid-sagittal and transabdominal plane. The probe was tilted caudally and posteriorly to obtain a clear image of the postero-inferior aspect (base) of the bladder, which varied depending on the bladder fullness of the participants.	2–5 MHz	The limit located between the hypoechoic region that represents the full bladder and the hyperechoic region that corresponds to the PF musculature.
LAW [33]	Linear	The right abdominal wall at the midpoint between the lower angle of the thoracic cage and the iliac crest. The medial edge of the transducer was placed approximately 10 mm from the alba linear.	7.5 MHz	Distance (mm) between the superficial and deep border of each muscle belly, which is marked by the hyperechoic fascial lines. The fascial lines were not included in the measurement.
LM [34]	Linear	L4–L5 facet joint over the articular pillar and the belly of the L4–L5 muscle over the articular pillar and the belly of the ML muscle.	5 MHz	Distance (mm) from the most posterior portion of the zygapophyseal joint of L4–L5 and the muscle surface separating it from the subcutaneous cellular tissue.

DPH: diaphragm; PF: pelvic floor; LAW: lateral abdominal wall; LM: lumbar multifidus.

**Table 3 diagnostics-14-02310-t003:** Summary of subject’s characteristics and baseline scores by allocation group.

		Cases (*n* = 26)	Controls (*n* = 34)	*p*	Total (*n* = 60)
Sex (Male/Female)	*n* (%)	12 (46.2)/14 (53.8)	17 (50)/17 (50)	0.768 a	29 (48)/31 (52)
Age (years)	Median (IQR)	27 (6)	25 (6)	0.475 b	26 (6)
Weight (kg)	Mean (SD)	69.47 (10.08)	65.71 (6.89)	0.092 c	67.34 (8.55)
Height (m)	Mean (SD)	1.70 (0.09)	1.70 (0.07)	0.964 c	1.70 (0.08)
BMI (kg/m^2^)	Mean (SD)	23.68 (1.90)	22.51 (1.92)	0.023 c *	23.02 (1.98)
Duration of pain (months)	Median (IQR)	2.95 (10)	-	-	
Pain episodes (continuous pain/1–5 episodes/>5 episodes)	*n* (%)	4 (15.4)/15 (57.7)/7 (26.9)	-	-	
Depression (No/Yes)	*n* (%)	20 (76.9)/6 (23.1)	-	-	
NSAID (No/Yes)	*n* (%)	18 (69.2)/8 (30.8)	-	-	
NSAID frequency (on demand/1 per month/>1 per month)	*n* (%)	4 (44.4)/3 (33.3)/2 (22.2)	-	-	
Oswestry	Median (IQR)	6 (12.5)	-	-	
VAS	Median (IQR)	31.5 (29.25)	-	-	

(a) Pearson’s χ^2^; (b) Mann–Whitney U test; (c) unpaired *t*-test; * *p* < 0.05. BMI: body mass index; NSAID: nonsteroidal anti-inflammatory drug; VAS: Visual Analog Scale.

**Table 4 diagnostics-14-02310-t004:** Activation patterns with higher frequency in each maneuver and position.

		Control (*n* = 34)	Case (*n* = 26)	*p*	E. Size
Standing—ADIM	*n* (%)			0.241 a	0.506 b
PF, LAW, DPH, LM		3 (8.8)	4 (15.4)		
LAW, DPH, PF, LM		6 (17.6)	0 (0)		
Standing—Valsalva	*n* (%)			0.241 a	0.506 b
DPH, PF, LAW, LM		6 (17.6)	0 (0)		
PF, LAW, DPH, LM		4 (11.8)	5 (19.2)		
LAW, DPH, PF, LM		0 (0)	5 (19.2)		
Standing—CAL	*n* (%)			0.241 a	0.506 b
LM, DPH, LAW, PF		4 (11.8)	0 (0)		
PF, LAW, DPH, LM		1 (2.9)	5 (19.2)		
Standing—VPFC	*n* (%)			0.144 a	0.555 b
LAW, PF, DPH, LM		8 (23.5)	2 (7.7)		
PF, LAW, LM PDH		3 (8.8)	4 (15.4)		
PF, DPH, LM, LAW		0 (0)	4 (15.4)		
Sitting—ADIM	*n* (%)			0.802 a	0.431 b
LAW, PF, DPH, LM		6 (17.6)	4 (15.4)		
Sitting—Valsalva	*n* (%)			0.241 a	0.506 b
LAW, PF, DPH, LM		4 (11.8)	1 (3.8)		
PF, LM, LAW, DPH		0 (0)	3 (11.5)		
DPH, PF, LAW, LM		0 (0)	3 (11.5)		
Sitting—CAL	*n* (%)			0.321 a	0.583 b
LAW, LM, DPH, PF		6 (17.6)	2 (7.7)		
DPH, LAW, LM, PF		2 (5.9)	5 (19.2)		
Sitting—VPFC	*n* (%)			0.308 a	0.532 b
PF, LAW, DPH, LM		6 (17.6)	4 (15.4)		
PF, LAW, LM, DPH		3 (8.8)	4 (15.4)		
LAW, PF, DPH, LM		1 (2.9)	4 (15.4)		

(a) Pearson’s χ2; (b) contingency coefficient. ADIM: abdominal drawing-in maneuver; CAL: contralateral arm lift; VPFC: voluntary pelvic floor contraction; PF: pelvic floor; LAW: lateral abdominal wall; DPH: diaphragm; LM: lumbar multifidus; E. Size: effect size.

## Data Availability

There is no more data available.

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
