# Peer review of "Ultrasound Evaluation of Onset Core Muscle Activity in Subjects with Non-Specific Lower Back Pain and Without Lower Back Pain: An Observational Case–Control Study"

_diagnostics, 2024, doi:10.3390/diagnostics14202310_

Round 1
Reviewer 1 Report
Comments and Suggestions for Authors
It is an interesting topic.
However, there are many aspects that need to be improved
Lines 34-46: ,, Keywords: Chronic Low Back Pain; Nonspecific Low Back Pain (NLBP); Rehabilitative Ultrasound Imaging (RUSI); Transversus Abdominis; Lumbar Multifidus; Diaphragm Function; Pelvic Floor Muscles; Muscle Contraction Ratios; Muscle Onset; Core Muscle Thickness”
There are too many keywords. I think 4 or 5 would have been enough.
Line 39: ,,Since 1990, low back pain (LBP) has been the leading cause of disability [1]. “
In Moissenet's paper, it is written that LBP is one of the 5 causes of disability, without specifying which one from the 5 is the main one.
Lines 198-200: ,,….both in sitting and standing positions. These positions were chosen because they are considered potential risk factors for the development and persistence of NSLBP [31].”
Are you referring to the fact that there is muscle hypotonia in the orthostatic position?
Lines 253-255: ,,The Participant Data Collection Form, which includes socio-demographic information such as height, weight, and personal medical history, was gathered from all participants.”
Why didn't you specify the body mass index and its role? IMC appears in Table no. 3 and in a comment in discussions, but the role of IMC does not appear in the text of Procedures.
Tabel 3 ,,Deepresion (No/Yes)”
Depression also appears in Table 3, but in the text you do not mention anything about this diagnosis, not the evaluation scale, only subjectively, probably a question to patients. Are you convinced that all patients have admitted that they have depression? Can depression somehow influence the contraction of the analyzed muscles? Even in the Discussions chapter, there is no statement if there is a correlation between changes in muscle thickness in patients with/without depression. So can you comment on why Depression appears in the table?
The objective is clear.
You mentioned the limits of the study. I think a larger number of participants is needed.
It was interesting to correlate these data obtained according to the gender of the participants and to see if these changes at the muscle level are different in female or male participants.
Further studies are certainly needed.
My comments are only intended to make the paper better. Good luck!
Author Response
COMMENTS 1: Lines 34-46: ,, Keywords: Chronic Low Back Pain; Nonspecific Low Back Pain (NLBP); Rehabilitative Ultrasound Imaging (RUSI); Transversus Abdominis; Lumbar Multifidus; Diaphragm Function; Pelvic Floor Muscles; Muscle Contraction Ratios; Muscle Onset; Core Muscle Thickness”
There are too many keywords. I think 4 or 5 would have been enough.
RESPONSE 1: Thank you for your recommendation. We have made the modification by reducing the number of keywords and selecting the following: Nonspecific Low Back Pain (NLBP); Rehabilitative Ultrasound Imaging (RUSI); Transversus Abdominis; Diaphragm Function; Pelvic Floor Muscles.
COMMENTS 2: Line 39: ,,Since 1990, low back pain (LBP) has been the leading cause of disability [1]. “
In Moissenet's paper, it is written that LBP is one of the 5 causes of disability, without specifying which one from the 5 is the main one.
RESPONSE 2: After reviewing the articles again, indeed, the reference by Moissenet mentions that low back pain has been the leading cause of disability worldwide since 1990. The appropriate bibliographic reference from which this data is derived is the article published in The Lancet (1).
- Global, regional, and national incidence, prevalence, and years lived with disability for 328 diseases and injuries for 195 countries, 1990-2016: a systematic analysis for the Global Burden of Disease Study 2016. Lancet (London, England). 2017 Sep;390(10100):1211–59.
COMMENTS 3: Lines 198-200: ,,….both in sitting and standing positions. These positions were chosen because they are considered potential risk factors for the development and persistence of NSLBP [31].”
Are you referring to the fact that there is muscle hypotonia in the orthostatic position?
RESPONSE 3: In this study, participants were asked to perform various maneuvers in both sitting and standing positions. Several studies have indicated that orthostatic positions (both sitting and standing) appear to be related to the development and prevalence of nonspecific chronic low back pain (2,3). Furthermore, we opted to conduct measurements in these positions due to their functional relevance, allowing us to observe simultaneous muscle behavior under load conditions, which can be translated to activities of daily living; therefore, we did not conduct measurements in the supine position.
- Inoue G, Miyagi M, Uchida K, Ishikawa T, Kamoda H, Eguchi Y, et al. Occupational Characteristics of Low Back Pain Among Standing Workers in a Japanese Manufacturing Company. J Orthop Sci Off J Japanese Orthop Assoc. 2020 Jan;68(1):23–30.
- Inoue G, Miyagi M, Uchida K, Ishikawa T, Kamoda H, Eguchi Y, et al. The prevalence and characteristics of low back pain among sitting workers in a Japanese manufacturing company. J Orthop Sci Off J Japanese Orthop Assoc. 2015 Jan;20(1):23–30.
COMMENTS 4: Lines 253-255: ,,The Participant Data Collection Form, which includes socio-demographic information such as height, weight, and personal medical history, was gathered from all participants.”
Why didn't you specify the body mass index and its role? IMC appears in Table no. 3 and in a comment in discussions, but the role of IMC does not appear in the text of Procedures.
RESPONSE 4: Thank you for your feedback. This must have been an oversight, which we have now corrected by including it in the procedures section (line 254).
COMMENTS 5: Tabel 3 ,,Deepresion (No/Yes)”
Depression also appears in Table 3, but in the text you do not mention anything about this diagnosis, not the evaluation scale, only subjectively, probably a question to patients. Are you convinced that all patients have admitted that they have depression? Can depression somehow influence the contraction of the analyzed muscles? Even in the Discussions chapter, there is no statement if there is a correlation between changes in muscle thickness in patients with/without depression. So can you comment on why Depression appears in the table?
RESPONSE 5: We appreciate your feedback. Several studies have linked depression to the perpetuation or as a causative factor of chronic nonspecific low back pain (4–6). Therefore, patients were asked about a history of depression to correlate it with the cases of low back pain and assess its possible involvement. However, after conducting the relevant statistical analyses, it was decided not to analyze this correlation in order to focus more specifically on the mechanical aspects of the pain process. Nevertheless, this variable was included in the table for readers who may be interested in this information.
For the assessment of depression, a question was included in the initial interview, but no validated diagnostic test or evaluation by a specialist physician was used. If it is deemed that this variable does not provide sufficient information, we are open to removing it. Again, we appreciate your feedback.
- Linton SJ, Bergbom S. Understanding the link between depression and pain. Scand J pain. 2011 Apr;2(2):47–54.
- Allegri M, Montella S, Salici F, Valente A, Marchesini M, Compagnone C, et al. Mechanisms of low back pain: a guide for diagnosis and therapy. F1000Research. 2016;5.
- Wong JJ, Tricco AC, Côté P, Liang CY, Lewis JA, Bouck Z, et al. Association Between Depressive Symptoms or Depression and Health Outcomes for Low Back Pain: a Systematic Review and Meta-analysis. Vol. 37, Journal of general internal medicine. United States; 2022. p. 1233–46.
COMMENTS 7: The objective is clear.
You mentioned the limits of the study. I think a larger number of participants is needed.
It was interesting to correlate these data obtained according to the gender of the participants and to see if these changes at the muscle level are different in female or male participants.
Further studies are certainly needed.
My comments are only intended to make the paper better. Good luck!
RESPONSE 7: We appreciate your comment. We agree that correlating the data based on participants' gender would be interesting and could provide additional insights into potential differences in muscle-level changes between male and female participants. However, the primary focus of the present study was on the mechanical aspects of low back pain, and a detailed analysis of gender differences in muscle data was not conducted. That being said, we believe this would be a relevant line of investigation for future studies and appreciate your suggestion.
Reviewer 2 Report
Comments and Suggestions for Authors
The subject of this manuscript is the evaluation of activity, and coordination of core muscles by the simultaneous collecting of ultrasound images in patients with CNSLBP and healthy subjects.
The introduction concerns the importance of the core muscles in developing lumbar spine complaints and how to assess them.
The section on materials, methods, and results is very clear and thoroughly describes the methodology of the research procedures.
The results section contains a description of the enrollment procedures (lines 321 -324).
*I ask the authors to present it in the form of a PRISMA diagram of the recruitment process in this section or in the materials and methods section.
Author Response
COMMENTS 1: *I ask the authors to present it in the form of a PRISMA diagram of the recruitment process in this section or in the materials and methods section.
RESPONSE 1:Thank you for your comments. As you suggested, we have included Figure 6, which contains the flow diagram, in the results section.
Reviewer 3 Report
Comments and Suggestions for Authors
Introduction
Line 39: What does LBP have to do with 1990? The disability is due to other incurable diseases of the musculoskeletal system, such as osteoarthritis of the knee, osteoarthritis of the hip, lumbar canal stenosis, inoperable herniated discs, etc.
In the introduction, the authors explain the link between nonspecific LBP and muscular imbalances of the Lumbar Multifidus, Lateral Abdominal Wall, Respiratory Diaphragm, and the Pelvic Floor.
Compared to healthy subjects, the study observes the onset of muscle contractions in subjects with nonspecific LBP. It also analyzes the relationship between rest and contraction/activation as well as between rest and excursion for various muscle groups. These variables may be essential because of their well-established link with lumbopelvic stability and core function. In addition, these factors are considered important predictors of pain and musculoskeletal function.
Materials and methods:
I thoroughly review the evaluations of healthy and ill subjects. The only mention of an assessment is in the selection criteria and is the VAS scale. How were the patients evaluated? Did they use specific clinical tests? Where do I see these tests?
Discussion and conclusion
While the subject matter is interesting, and the technology used, the results of the study do not provide significant novelty, as it did not identify a specific pattern of muscle activation among subjects with chronic nonspecific LBP compared to healthy subjects. Although variations in muscle thickness and excursion were observed in certain maneuvers, these are not sufficient to establish a clear pattern or direct link to low back pain. This suggests that the mechanisms of chronic low back pain remain largely unknown and multifactorial, requiring further research.
Although I respect the researchers' efforts, unfortunately, this article does not meet the publication standards required for this journal. That said, the methodology employed is highly interesting and has potential for creating valuable correlations between clinical and imaging findings in specific pathologies
Author Response
COMMENTS 1: Line 39: What does LBP have to do with 1990? The disability is due to other incurable diseases of the musculoskeletal system, such as osteoarthritis of the knee, osteoarthritis of the hip, lumbar canal stenosis, inoperable herniated discs, etc.
RESPONSE 1: It is true that low back pain (LBP) is not the only cause of disability related to incurable musculoskeletal conditions. However, low back pain is a highly relevant factor in global disability, as it affects a large number of people and has a significant impact on quality of life, both due to its prevalence and chronic nature. According to the article published in The Lancet in 2017 (1), low back pain has been classified as the leading cause of disability since 1990, not only because of its frequency but also due to the difficulty in its treatment and management. The mention of the year was intended to highlight the importance and relevance of this issue. However, if you feel that this reference is not essential, we are happy to modify it immediately to align with your suggestions
- Global, regional, and national incidence, prevalence, and years lived with disability for 328 diseases and injuries for 195 countries, 1990-2016: a systematic analysis for the Global Burden of Disease Study 2016. Lancet (London, England). 2017 Sep;390(10100):1211–59.
COMMENTS 2: In the introduction, the authors explain the link between nonspecific LBP and muscular imbalances of the Lumbar Multifidus, Lateral Abdominal Wall, Respiratory Diaphragm, and the Pelvic Floor.
Compared to healthy subjects, the study observes the onset of muscle contractions in subjects with nonspecific LBP. It also analyzes the relationship between rest and contraction/activation as well as between rest and excursion for various muscle groups. These variables may be essential because of their well-established link with lumbopelvic stability and core function. In addition, these factors are considered important predictors of pain and musculoskeletal function.
Materials and methods:
I thoroughly review the evaluations of healthy and ill subjects. The only mention of an assessment is in the selection criteria and is the VAS scale. How were the patients evaluated? Did they use specific clinical tests? Where do I see these tests?
RESPONSE 2: First of all, we would like to thank you for your comments. Participants for the chronic non-specific low back pain group were required to have a diagnosis of nonspecific low back pain confirmed by a primary care physician. Additionally, the pain had to have lasted more than 12 weeks within the past year, and at the time of study inclusion, subjects had to score higher than 3 on the visual analogue scale. If these criteria were met and subjects were within the age range of 18 to 60 years, they were eligible for selection for the study. Exclusion criteria included pregnancy, general surgical intervention, vertebral pathology (such as fractures, cancer, or infection), inability to stand or sit, or inability to perform the required maneuvers.
COMMENTS 3: Discussion and conclusion
While the subject matter is interesting, and the technology used, the results of the study do not provide significant novelty, as it did not identify a specific pattern of muscle activation among subjects with chronic nonspecific LBP compared to healthy subjects. Although variations in muscle thickness and excursion were observed in certain maneuvers, these are not sufficient to establish a clear pattern or direct link to low back pain. This suggests that the mechanisms of chronic low back pain remain largely unknown and multifactorial, requiring further research.
Although I respect the researchers' efforts, unfortunately, this article does not meet the publication standards required for this journal. That said, the methodology employed is highly interesting and has potential for creating valuable correlations between clinical and imaging findings in specific pathologies
RESPONSE 3: We greatly appreciate your feedback. We would like to highlight the novelty of this study, as we have successfully developed a specific methodology to simultaneously measure four abdominopelvic muscle groups, which are considered highly relevant in their involvement with nonspecific chronic low back pain. To the best of our knowledge, after a thorough review of the literature, we did not find previous studies that had addressed the same objective as the present study, or perhaps such studies existed but did not find the appropriate method to carry it out. This study presents a new method for the simultaneous measurement of different muscle groups in the abdomen and pelvic floor using ultrasound.
On the other hand, a significant finding in this study was the identification of 41 different muscle activation patterns, highlighting the considerable heterogeneity within the population. The fact that no specific pattern of simultaneous muscle activation was found also represents an important finding, emphasizing the need to educate motor control patterns in an individualized manner, based on the assessment conducted. These results are consistent with the existing literature, where several authors note the significant individual variations in deep abdominal muscle activation in patients with chronic low back pain(7), while others highlight the highly adaptable nature of the motor response, suggesting that interventions should be individualized in treatments (8)
We would also like to stress the importance of publishing study results, even when they are not as expected, in order to avoid publication bias (9).
Finally, we believe that this study could provide new insights for diagnostic assessment in subjects for whom the study of simultaneous muscle contraction is relevant. Observing muscle behavior in this way could be an interesting approach for future studies and research directions
- Vasseljen O, Unsgaard-Tøndel M, Westad C, Mork PJ. Effect of core stability exercises on feed-forward activation of deep abdominal muscles in chronic low back pain: a randomized controlled trial. Spine (Phila Pa 1976). 2012 Jun;37(13):1101–8.
- Merkle SL, Sluka KA, Frey-Law LA. The interaction between pain and movement. J hand Ther Off J Am Soc Hand Ther. 2020;33(1):60–6.
- Marks-Anglin A, Chen Y. A historical review of publication bias. Res Synth Methods. 2020 Nov;11(6):725–42.
Round 2
Reviewer 3 Report
Comments and Suggestions for Authors
The team of authors successfully made the necessary revisions that led to a rejection. The way they addressed and scientifically substantiated my comments makes me fully accept this improved version